# Longitudinal Assessment of Quality of Life in Nasopharyngeal Cancer Patients Treated with Intensity-Modulated Proton Therapy and Volumetric Modulated Arc Therapy at Different Time Points

**DOI:** 10.3390/cancers16061217

**Published:** 2024-03-20

**Authors:** Kuan-Cho Liao, Yu-Jie Huang, Wen-Ling Tsai, Chien-Hung Lee, Fu-Min Fang

**Affiliations:** 1Department of Radiation Oncology, Kaohsiung Chang-Gung Memorial Hospital and Chang Gung University College of Medicine, Kaohsiung 833401, Taiwan; piko@cgmh.org.tw (K.-C.L.); yjhuang@cgmh.org.tw (Y.-J.H.); 2Department of Public Health, College of Health Sciences, Kaohsiung Medical University, Kaohsiung 807378, Taiwan; 3Department of Cosmetics and Fashion Styling, Center for Environmental Toxin and Emerging-Contaminant Research, Cheng Shiu University, Kaohsiung 833301, Taiwan; k0605@gcloud.csu.edu.tw; 4Research Center for Environmental Medicine, Kaohsiung Medical University, Kaohsiung 807378, Taiwan; 5Department of Medical Research, Kaohsiung Medical University Hospital, Kaohsiung Medical University, Kaohsiung 807378, Taiwan; 6Office of Institutional Research & Planning, Secretariat, Kaohsiung Medical University, Kaohsiung 807378, Taiwan; 7Department of Medicine, College of Medicine, Chang Gung University, Taoyuan 333323, Taiwan

**Keywords:** nasopharyngeal cancer, intensity-modulated proton therapy (IMPT), volumetric modulated arc therapy (VMAT), quality of life, interaction effect, time-dependent effect

## Abstract

**Simple Summary:**

Intensity-modulated proton therapy (IMPT) has proven to be more effective in minimizing radiation exposure to normal organs when compared to photon-based volumetric modulated arc therapy (VMAT) in patients diagnosed with nasopharyngeal cancer (NPC). This retrospective cohort study represents the first quantitative assessment of the longitudinal impact on the quality of life (QoL) outcomes in NPC patients undergoing treatment with IMPT (*n* = 41) as opposed to VMAT (*n* = 246). We gathered data on global QoL, functional QoL, C30 symptoms, and HN35 symptoms using the EORTC QLQ-C30 and QLQ-HN35 questionnaires at four time points: pre radiotherapy (RT), during RT, 3 months post RT, and 12 months post RT. IMPT demonstrated superior mean dose reductions in 12 of the 16 organs at risk compared to VMAT. This reduction in radiation dose, attributed to the IMPT technique, appears to be associated with positive outcomes in functional QoL, reflecting a noteworthy increase of 7.5 points and a decrease of 10.7 points in HN35 symptoms. However, this effect is time dependent and exclusively observed at the time point of during RT.

**Abstract:**

Purpose: This retrospective cohort study aims to compare the quality of life (QoL) in patients with nasopharyngeal cancer (NPC) treated with intensity-modulated proton therapy (IMPT) versus volumetric modulated arc therapy (VMAT) at different time points. Materials and Methods: We conducted a longitudinal assessment of QoL on 287 newly diagnosed NPC patients (IMPT: 41 and VMAT: 246). We collected outcomes of global QoL, functional QoL, C30 symptoms, and HN35 symptoms from EORTC QLQ-C30 and QLQ-HN35 questionnaires at pre-radiotherapy, during radiotherapy (around 40 Gy), 3 months post radiotherapy, and 12-months post radiotherapy (RT). The generalized estimating equation was utilized to interpret the group effect, originating from inherent group differences; time effect, attributed to RT effects over time; and interaction of the group and time effect. Results: IMPT demonstrated superior mean dose reductions in 12 of the 16 organs at risk compared to VMAT, including a significant (>50%) reduction in the oral cavity and larynx. Both groups exhibited improved scores of global QoL, functional QoL, and C30 symptoms at 12 months post RT compared to the pre-RT status. Regarding global QoL and C30 symptoms, there was no interaction effect of group over time. In contrast, significant interaction effects were observed on functional QoL (*p* = 0.040) and HN35 symptoms (*p* = 0.004) during RT, where IMPT created an average of 7.5 points higher functional QoL and 10.7 points lower HN35 symptoms than VMAT. Conclusions: Compared to VMAT, dose reduction attributed to IMPT could translate into better functional QoL and HN35 symptoms, but the effect is time dependent and exclusively observed during the RT phase.

## 1. Introduction

Radiotherapy (RT) and chemotherapy stand as the primary treatment modalities for individuals diagnosed with nasopharyngeal carcinoma (NPC). Volumetric modulated arc therapy (VMAT), an advanced form of photon beam therapy that utilizes the intensity-modulated RT (IMRT) technique through rotational beam delivery, has become a widely employed RT technique for treating NPC [1]. On the other hand, intensity-modulated proton therapy (IMPT) represents a specialized approach within proton beam therapy. IMPT utilizes the inherent physical properties of the Bragg peak to concentrate the maximum radiation dose precisely on the tumor target, minimizing the radiation dose beyond the target area. The adoption of IMPT as a treatment modality for NPC has been progressively increasing in medical institutions globally, owing to its recognized clinical advantages [2].

Assessing the quality of life (QoL) of patients with NPC at different stages of clinical treatment provides valuable insights into the impact of interventions on their physical, emotional, and social well-being. Such evaluations contribute to a deeper understanding of how diverse clinical approaches influence the overall well-being of patients, aiding informed decision-making and the optimization of treatment strategies [3]. Compared to earlier two-dimensional or three-dimensional conformal RT techniques, IMRT or VMAT has demonstrated an improvement in the QoL of NPC patients by mitigating side effects such as dry mouth and sticky saliva [4,5,6,7].

IMPT has been evidenced to improve the QoL of patients diagnosed with brain, head and neck, lung, and pediatric cancers [8]. Furthermore, various studies have established the effectiveness of IMPT in reducing both acute and late toxicity in patients with NPC [2,9,10,11,12,13]. However, there is a paucity of research that directly compares the impact on QoL between NPC patients undergoing IMPT and those undergoing VMAT. This study aims to address this gap by longitudinally comparing the QoL outcomes for NPC patients undergoing IMPT versus VMAT at different time points.

## 2. Patients and Methods

### 2.1. Patient Cohort

Kaohsiung Chang Gung Memorial Hospital (KCGMH) initiated the use of VMAT in treating NPC patients in January 2011 and adopted IMPT in January 2019. This retrospective cohort study was conducted at the Department of Radiation Oncology in KCGMH, Taiwan. NPC patients were recruited from outpatient visits, where the treatment plan and RT schedule were determined. We recruited 287 patients with pathology-confirmed, newly diagnosed, and non-metastatic NPC at KCGMH, spanning the period from January 2011 to September 2022. The inclusion criteria encompassed patients who received either IMPT or VMAT for the entire treatment course and those who completed the prescribed QoL questionnaires. Individuals with a history of other cancers or prior RT and/or chemotherapy were excluded from the study.

### 2.2. Treatment

The technique of VMAT and IMPT for NPC patients in this institute was published in detail previously [14,15]. The regimens of target delineation, dose prescription, and fractionation were consistent for VMAT and IMPT patients, with 69.96 Gy, 59.4 Gy, and 52.8–54.0 Gy in 33 fractions, 1 fraction per day and 5 fractions per week, at the high, middle, and low dose levels of clinical target volume (CTV-H, CTV-M, and CTV-L), respectively.

The CTV-H was determined to encompass the gross tumor and associated nodes, extending isotropically by 3 mm from the boundaries of the gross tumor volume (GTV) as visualized in the imaging studies. The CTV-M was delineated to encompass neighboring anatomical structures at risk, such as the skull base, parapharyngeal space, and upper neck lymphatics, to cover potential routes of micro-metastasis from the disease. The CTV-L was designed to include the subclinical lymphatics in the lower neck region not directly involved with the tumor. The organs at risk (OARs) were contoured with specified dose constraints to minimize radiation exposure and reduce treatment-related side effects. These OARs included critical structures such as the brain, brainstem, spinal cord, optic nerve, lens, optic chiasm, cochleas, parotid glands, submandibular glands, oral cavity, mandible, larynx, and thyroid gland, among others. The applied constraints on these OARs typically followed established guidelines and recommendations to ensure optimal treatment outcomes while minimizing adverse effects [16]. The IMPT group utilized the scanning beam technique with a Sumitomo Proton Machine for treatment delivery. Treatment planning was performed using the RayStation treatment planning system (version 7, Raysearch Medical Laboratories, Stockholm, Sweden). Typically, planning involved utilizing three-beam directions: left and right anterior obliques, along with posterior fields, utilizing multi-field optimization. Robust optimization was generally employed to account for range uncertainties (plus 3.5%) and positional uncertainties (plus 3 mm). Robust evaluation, involving the creation of 21 plans ranging from worst- to best-case scenarios, was utilized to assess the plans. Daily CT-based image guidance was conducted to ensure setup accuracy.

For the VMAT group, the Philips Pinnacle Planning System version 9.2 (Philips, Fitchburg, WI, USA) was used. The planning strategy comprised dual coplanar arcs covering 360°, including both clockwise and counterclockwise directions. The planning target volume (PTV) was generated with additional margins of 3–5 mm around each CTV. Each treatment plan underwent a thorough evaluation to guarantee that 95% of all PTVs received the prescribed dose. Quantitative assessment of dose volume histograms for all of the target volumes and OAR was conducted, while qualitative inspection involved scrutinizing the isodose curves on axial CT slices for each IMPT or VMAT plan.

In most cases, for patients with clinical stages III–IVA, neoadjuvant chemotherapy was given intravenously. The combination regimens included cisplatin (60–75 mg/m^2^, day 1) plus gemcitabine (1 g/m^2^, day 1 and 8) or cisplatin (70–80 mg/m^2^ on day 1) plus 5-fluorouracil (700–800 mg/m^2^/day on day 1–4) every 3 weeks per cycle for 3 cycles [17,18]. During the treatment course of IMPT or VMAT for individuals with clinical stages II–IVA, concurrent chemotherapy was administered. This involved weekly intravenous cisplatin at a dosage of 40 mg/m^2^ for a duration of 6 or 7 weeks, serving as a radiation sensitizer [16].

### 2.3. QoL Instruments

The QoL data from each patient were gathered using the Taiwan Chinese version 3 of core QoL questionnaire for cancer patients (QLQ-C30) and the head and neck cancer-specific QoL questionnaire module (QLQ-HN35), both developed by the European Organization for Research and Treatment of Cancer (EORTC) based in Brussels, Belgium [19]. The EORTC QLQ-C30 is an extensive tool for assessing QoL, comprising a global QoL scale, five functional scales covering physical, cognitive, role, emotional, and social functioning, as well as nine scales/items addressing cancer symptoms such as fatigue, nausea and vomiting, pain, dyspnea, insomnia, diarrhea, constipation, appetite loss, and financial difficulties [20]. EORTC QLQ-HN35 includes seven symptom problem scales (pain, swallowing, speech, senses, social contact, social eating, and sexuality), six symptom problem items (opening mouth trouble, teeth, sticky saliva, dry mouth, coughing, and feeling ill), and five dichotomous items (use of painkillers, nutritional supplements, feeding tube, weight loss, and weight gain).

In the QoL questionnaires, all scales and items were structured with a four-point Likert scale, except for five dichotomous items in the QLQ-HN35 questionnaire and one seven-point Likert scale for global health in the QLQ-C30 questionnaire. The responses were linearly converted to a score between 0 and 100 [20]. To evaluate the characteristic-specific influence and avoid potential type I error derived from multiple tests, we respectively calculated the average QoL scores of the scales/items in the groups of “functional QoL”, “C30 symptoms”, and “HN35 symptoms”. In EORTC QLQ-C30, the average QoL scores for “functional QoL” and “C30 symptoms” characteristics were computed by five functional scales and nine general cancer symptom-related scales/items, respectively. In EORTC QLQ-HN35, the average QoL score for “HN35 symptoms” was evaluated by eighteen head and neck cancer-related symptoms. In the QoL questionnaires, a high QoL score implies a better global QoL and functional QoL, or more symptoms or problems disorder.

### 2.4. Time Points of QoL Assessment

In this NPC patient cohort, participants were evaluated for their QLQ-C30 and QLQ-HN35-derived QoL before receiving RT (pre-RT, baseline). The conditions of QoL were also assessed for each patient during RT when the dose reached around 40 Gy and at 3 and 12 months post RT. The pre-RT period was defined as the duration from the patient’s agreement to participate in the study during the outpatient visit to the commencement of their first RT fraction, which occurred approximately 1–2 weeks later. The assessments at the four time points represent the clinical information at baseline and the phases of acute, subacute, and late RT effects, respectively.

### 2.5. Covariates

We collected demographic factors and clinical variables for each participant at the first assessment for patient clinical conditions. Age was categorized into three groups: ≤40, 41–65, and >65 years. Ethnicity was categorized into Minnan and non-Minnan groups, with the latter including Mainlander, Hakka, and Aborigines. Educational level was divided into two categories: ≤12 years and >12 years. Body mass index (BMI) was classified into two groups: non-overweight (BMI < 24) and overweight/obesity (BMI ≥ 24), based on the criteria established by the Taiwan Health Promotion Administration [21]. All NPC patient staging was determined using the American Joint Committee on Cancer (AJCC) Staging Systems 8th edition, with patients divided into stages I–II and III–IVA [22]. The history of chronic disease was defined as patients having at least one of the following diseases: diabetes, heart disease, hypertension, stroke, arthritis, asthma, tuberculosis, peptic ulcer disease, chronic hepatitis, and liver cirrhosis.

### 2.6. Statistical Analysis

Proportions and means with standard deviations were used to describe the distributions of categorical and continuous study parameters, respectively, for NPC patients treated with VMAT and IMPT. Age, sex, ethnicity, educational level, BMI, chronic disease, AJCC stage, and chemotherapy were considered as potential confounders and were accounted for in the assessment of effects in all multivariable models. Multiple linear regression models were utilized to assess the differences in QoL score between patients receiving VMAT and IMPT treatments at different time points [23]. Additionally, generalized estimating equations (GEEs) with an autoregressive correlation structure were employed to interpret the group effect, originating from inherent group differences; time effect, attributed to RT effects over time; and interaction between RT groups and time points on QoL outcomes over a 12-month follow-up period. GEE is a robust statistical method specifically designed for analyzing longitudinal or clustered data, where observations within clusters are correlated, such as repeated measurements on individuals [24]. It extends the generalized linear model framework and can handle various data types and correlation structures. GEE provides consistent parameter estimates, even when the correlation structure is misspecified, through an iterative estimation procedure. A *p*-value < 0.05 was considered statistically significant in all tests.

## 3. Results

### 3.1. Demographic and Clinical Characteristics

This study encompassed a cohort of 287 NPC patients, with 85.7% (*n* = 246) receiving VMAT and 14.3% (*n* = 41) receiving IMPT (Table 1). The two groups of NPC patients exhibited comparable demographic characteristics, including age, sex, ethnicity, educational level, and BMI. However, the IMPT group had a higher prevalence of chronic diseases compared to the VMAT group (56.1% vs. 32.7%, *p* = 0.004). There were no significant differences between the two RT groups in terms of AJCC staging and chemotherapy treatment.

### 3.2. Radiation Dose for Organs at Risk

The mean doses for OAR ranged from 577.0 ± 91.6 to 4165.2 ± 675.7 cGy for the VMAT group and 285.1 ± 160.4 to 3694.0 ± 275.0 cGy for the IMPT group (Table 2). Except for the optic nerves and parotid glands, IMPT delivered significantly lower mean doses in 12 out of the 16 OARs compared to VMAT. Notably, the IMPT group exhibited a more than 50% reduction in mean dose for structures such as the oral cavity, larynx, mandible, and lens.

### 3.3. QLQ-C30 QoL Scores for VMAT versus IMPT

The return rates for the QoL investigation at the four time points were as follows: 100% (*n* = 287) for pre-RT, 86.8% (*n* = 249) during RT, 84.7% (*n* = 243) at 3 months post RT, and 73.9% (*n* = 212) at 1 year post RT. The covariate-adjusted QoL scores for the QLQ-C30 scales in the VMAT and IMPT groups at the four time points are detailed in Table 3. Initially, at baseline, no significant difference was observed in most of the QLQ-C30 scales, except for cognitive function (aDiff., 6.0). During RT, the IMPT group exhibited a significantly higher global QoL score (aDiff., 14.6) along with elevated scores in physical, cognitive, and social functioning (aDiff., 7.8 to 9.7) and lower scores in pain and financial difficulties (aDiff., −10.3 and −15.8, respectively). At 3 months post RT, the IMPT group also displayed reduced scores in diarrhea and financial difficulties (aDiff., −7.6 and −16.6, respectively). By 12 months post RT, the IMPT group demonstrated higher scores in physical, emotional, cognitive, and social functioning (aDiff., 5.3 to 10.3) and lower scores in fatigue, nausea and vomiting, appetite loss, and financial difficulties (aDiff., −4.5 to −14.9).

### 3.4. QLQ-HN35 QoL Scores for VMAT versus IMPT

Table 4 displays the adjusted QoL scores of the EORTC QLQ-HN35 scales for NPC patients undergoing VMAT and IMPT at different time points. Initially, at baseline, the IMPT group exhibited significant lower scores in 7 of the 18 items (aDiff., −6.2 to −21.4) and higher scores in 2 of the 18 items (aDiff., 19.8 to 23.7). During RT, the IMPT group showed lower QoL scores in nine items including pain, swallowing, speech problems, trouble with social eating, teeth, opening mouth, coughing, painkillers, and weight loss compared to the VMAT group (aDiff., −9.6 to −33.7). A similar pattern persisted at 3 months post RT (aDiff., −6.0 to −17.8), except for sticky saliva, where the IMPT group had a higher score (aDiff., 16.4). At 12 months post RT, the IMPT group demonstrated reduced scores on 11 items of HN35 symptoms (aDiff., −5.8 to −19.0).

### 3.5. Longitudinal Changes of QoL between RT Groups

Table 5 presents the group effect, time effect, and interaction effect between RT groups and time points on average scores of QLQ-C30 and QLQ-HN35 QoL, while Figure 1 illustrates the longitudinal changes in score. A statistically significant group effect was observed in global QoL and C30 symptoms but not in functional QoL or HN35 symptoms. This indicates inherent differences between the IMPT group and VMAT group in global QoL and C30 symptoms throughout the four time points (Figure 1A,C). Conversely, a statistically significant time effect was observed in all the four QoL outcomes. Compared to pre-RT status, the study patients experienced a statistically significant decrease in global QoL (score, −15.3) and functional QoL (score, −8.4), along with an increase in C30 symptoms (score, 14.5) and HN35 symptoms (score, 24.6) during RT. However, these QoL outcome showed improvement at 3 months post RT, with significantly better outcome observed at 12 months post RT than those at pre-RT status.

The statistical analysis demonstrated a significant interaction effect between RT groups and time points, particularly in functional QoL and HN35 symptoms. However, no such interaction effect was observed in global QoL and C30 symptoms. Specifically, the interaction effect revealed a notably higher score in functional QoL for the IMPT group compared to the VMAT group during RT (82.7 vs. 75.2, ΔDiff. = 7.5, *p* for interaction = 0.040, Figure 1B). Additionally, a lower score in HN35 symptoms was observed in the IMPT group compared to the VMAT group during RT (30.1 vs. 40.8, ΔDiff. = −10.7, *p* for interaction = 0.004, Figure 1D). These findings highlight significant differences between the two RT groups in terms of functional QoL and HN35 symptoms during RT.

## 4. Discussion

Based on an extensive literature review, our study represents the first quantitative evaluation of longitudinal QoL outcomes in NPC patients undergoing treatment with VMAT versus IMPT. The most noteworthy result in this investigation is the substantial decrease in mean radiation dose across most observed OARs in patients treated with IMPT compared to VMAT. This dose reduction, attributed to the IMPT technique, appears to correlate with favorable outcomes in functioning QoL and HN35 symptoms. However, it is essential to emphasize that this effect is time-dependent, manifesting exclusively during the RT phase.

Analyzing our dataset, IMPT exhibited superior mean dose reductions in 12 out of 16 identified OARs compared to VMAT. A similar trend was noted in comparison to IMRT by Lewis et al., with a significantly lower mean dose to 13 OARs in a proton-based plan for patients with NPC compared with IMRT [25]. This superiority included a noteworthy reduction (>50%) in specific midline mucosal structures, such as the oral cavity and larynx. The observed dose reduction in these radiation-sensitive organs may elucidate the therapeutic advantage of IMPT over VMAT during RT. In a comparison of IMPT and IMRT for NPC patients, Li et al. also observed significant mean dose reductions in the oral cavity (IMPT: 15.4 Gy vs. IMRT: 32.8 Gy) and larynx (IMPT: 16.0 Gy vs. IMRT: 29.6 Gy) [13]. Their study revealed a significant trend toward lower grades of acute side effects in the IMPT group, encompassing dysphagia, fatigue, xerostomia, dysgeusia, oral mucositis, weight loss, and hoarseness.

In contrast to 2D-RT or 3D-CRT, parotid-sparing IMRT/VMAT has demonstrated efficacy in diminishing both parotid dose and the occurrence and severity of xerostomia for NPC patients [26,27,28]. Nevertheless, our study, alongside others, did not reveal a significant reduction in parotid dose through IMPT when compared to IMRT/VMAT [9,25]. Lewis et al. and Jakobi et al. identified patients with tumors located in the upper region of the head and neck as those who stand to derive the most substantial benefits from IMPT, particularly in terms of mitigating swallowing-related side effects. The pharyngeal constrictor muscles were regarded to be a critical structure in preventing dysphagia-related symptoms in patients receiving head and neck irradiation [29]. The dose to the pharyngeal constrictor muscles were not regularly constrained in the study cohort, and as a result, their contribution to the HN35 symptoms could not be specified.

According to reports by King and Osoba et al., a 10-point deviation on a single scale within 0–100 range was deemed clinically significant in QoL comparisons between groups [30]. Notably, there are no established reference scores indicating clinical significance when comparing the average scores of groups on various scales or items. It is essential to recognize that a statistical difference does not inherently imply clinical significance, and vice versa, as factors such as small sample size or non-random choice can influence results [30,31]. Interpreting the clinical relevance of a 5.0-point increase in functional QoL and a 7.1-point decrease in HN35 symptoms is challenging, given the absence of clear benchmarks, making it uncertain as to whether these differences are practically meaningful in a clinical context.

The QoL trajectory for VMT- or IMPT-treated NPC patients exhibited a similar pattern to our prior investigation involving patients treated with IMRT [26,32]. A systematic assessment comprising 14 studies, focusing on QoL in NPC patients treated with IMRT, indicated that the worst QoL outcomes, as measured by multidimensional scales, were observed during or at the end of treatment. However, a gradual recovery in QoL was evident within 1–2 years post IMRT [33]. Notably, the observed improvements extend beyond the immediate post-RT period, indicating a sustained recovery in various QoL aspects. It is worth emphasizing the importance of continued monitoring and support for patients during the recovery phase, especially given the lingering symptoms at 12 months post RT.

A time-dependent impact on longitudinal QoL outcomes between two comparative RT techniques has been identified. In our prior investigation comparing 3D-CRT with IMRT for treating NPC, it was found that the potential advantage of IMRT over 3D-CRT in QoL outcomes was evident solely during the recovery phase of acute toxicity [26]. Similarly, Sio et al., comparing IMPT with IMRT for patients with oropharyngeal carcinoma, observed that, according to the MD Anderson Symptom Inventory for Head and Neck Cancer module, symptom burden was lower among the IMPT patients than among the IMRT patients during the subacute recovery phase after treatment [34]. In the current study, the therapeutic advantage of IMPT over VMAT was notably evident at the acute phase. Nevertheless, a subtle divergence in trends between IMPT and VMAT emerged in functional QoL and HN35 symptoms from 3 to 12 months post RT. If this trend in QoL beyond 1 year post RT continues, we anticipate that IMPT technology may offer better preservation of long-term QoL at the late phase. However, this hypothesis still requires further data collection and research on long-term QoL to substantiate it.

The debate persists on whether the dosimetric outcomes associated with advanced RT techniques translates into global QoL or specific functional QoL improvements. Notably, in the context of treating head and neck cancer, a significant reduction in parotid dose (27 Gy vs. 43 Gy) with IMRT compared to 3D-CRT resulted in a marked decrease in patient- and observer-rated xerostomia, along with other head and neck symptoms. This reduction significantly correlated with improvements in the broader dimensions of QoL [35]. Conversely, despite a substantial reduction (25–30%) in the mean dose to normal structures such as the parotid glands and oral cavity with IMRT versus 3D-CRT, this did not manifest as a measurable enhancement in global or functional QoL [26]. Beyond dosimetric outcomes at OARs, various sociodemographic factors may introduce bias into the interpretation of global or functional QoL outcomes between comparative RT groups [36,37]. Moreover, alterations in QoL following treatment reflect the combined impacts of tumor regression and perceived complications from the administered treatment. Pre RT, patients grapple with the recent, often distressing experience of a cancer diagnosis, coupled with the uncertainty of a potentially life-threatening treatment. Post treatment, as patients adapt to the situation and experience tumor shrinkage, recovery from acute side effects, and lingering late complications in the head and neck area, they undergo a complex process of reappraising life domains, reviewing life goals, and adjusting satisfaction with life themes. An influential factor in this adaptation process, termed ‘response shift’, further complicates the interpretation of changes in QoL data over time [38].

This study is subject to several limitations. Firstly, the relatively low number of NPC patients treated with IMPT in our sample is noteworthy. This can be attributed to the elevated medical costs associated with IMPT and its lack of coverage by general insurance in the country. Second, the potential impact of socioeconomic factors on QoL poses a consideration. While control over socioeconomic parameters is constrained, this study has attempted to address the effects of age, sex, ethnicity, educational level, and body mass index on the reported QoL among patients treated with both VMAT and IMPT. Third, IMPT-treated NPC patients showed higher cognitive function compared to VMAT-treated patients. In Taiwan, where IMPT is not covered by insurance, IMPT patients tend to have higher socioeconomic status. Higher socioeconomic status is associated with better cognitive function [39,40], explaining our study’s findings. We had accounted for education level as a proxy for socioeconomic status to address this issue. Last, the availability of follow-up data for patients treated with IMPT was limited to a relatively short duration. This limitation stems from the recent introduction of IMPT at the institution. Consequently, this study can only offer insights into the treatment outcomes related to QoL observed 12 months post IMPT.

## 5. Conclusions

Compared to VMAT, dose reduction attributed to IMPT could translate to better functional QoL and HN35 symptoms, but the effect is time dependent and exclusively observed during the RT phase.

## Figures and Tables

**Figure 1 cancers-16-01217-f001:**
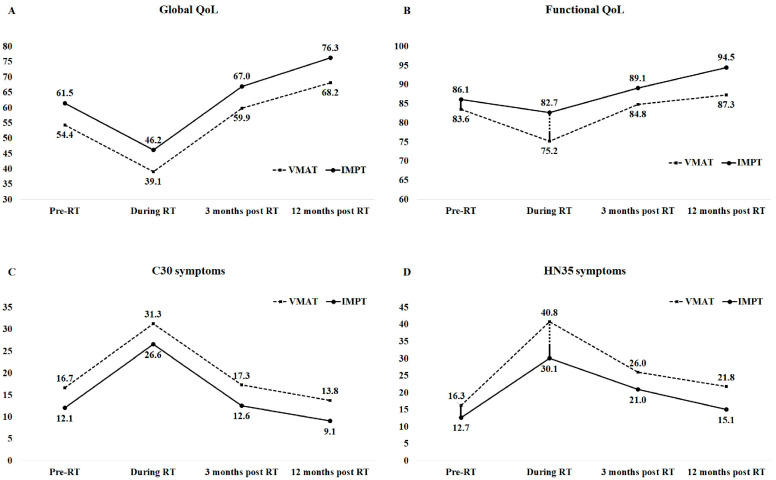
The benefit of interaction effects of radiation therapy technique and clinical time points on average scores for (**A**) global quality of life (QoL), (**B**) functional QoL, (**C**) C30 symptoms, and (**D**) HN35 symptoms. RT, radiotherapy; VMAT: volumetric modulated arc therapy; IMPT: intensity-modulated proton therapy. The scores were the QoL average scores obtained from EORTC QLQ-C30 5 functional scales, 9 C30 symptoms, and 18 QLQ-HN35 symptoms or items. The solid black line in the Pre-RT and During RT represents the main effects of IMPT. The dashed black line during RT indicates the interaction effects of the IMPT technique at the clinical time points.

**Table 1 cancers-16-01217-t001:** Distributions of demographic and clinical factors in nasopharyngeal cancer patients with VMAT and IMPT.

Factors	All (*n* = 287)		VMAT (*n* = 246)		IMPT (*n* = 41)		*p*-Value ^a^
	No.	%	No.	%	No.	%	
Age							0.123
≤40	54	18.8	50	20.3	4	9.8	
41–65	199	69.4	165	67.1	34	82.9	
>65	34	11.8	31	12.6	3	7.3	
Sex							0.097
Male	206	71.8	181	73.6	25	61.0	
Female	81	28.2	65	26.4	16	39.0	
Ethnicity							0.090
Minnan	240	83.6	202	82.1	38	92.7	
Non-Minnan	47	16.4	44	17.9	3	7.3	
Educational level							0.458
≤12 years	203	70.7	172	69.9	31	75.6	
>12 years	84	29.3	74	30.1	10	24.4	
Body mass index, kg/m^2 b^							0.339
Non-overweight (<24)	115	40.9	101	42.1	14	34.1	
Overweight (≥24)	166	59.1	139	57.9	27	65.9	
Chronic disease ^c^							0.004
No	183	64.0	165	67.3	18	43.9	
Yes	103	36.0	80	32.7	23	56.1	
AJCC stage							0.959
I–II	97	33.8	83	33.7	14	34.1	
III–IV	190	66.2	168	66.3	27	65.9	
T stage							0.785
T1–T2	190	66.4	162	66.1	28	68.3	
T3–T4	96	33.6	83	33.9	13	31.7	
N stage							0.311
N0–N1	147	51.2	123	50.0	24	58.5	
N2–N3	140	48.8	123	50.0	17	41.5	
Chemotherapy							0.300
No	19	6.6	14	5.7	5	12.2	
Concurrent	67	23.3	58	23.6	9	21.9	
Induction + Concurrent	201	70.1	174	70.7	27	65.9	

VMAT: volumetric modulated arc therapy; IMPT: intensity-modulated proton therapy; AJCC: American Joint Committee on Cancer8th edition. ^a^
*p*-values were obtained from chi-squared tests or Fisher’s exact tests. ^b^ Body mass index groups were categorized according to the criterion of Health Promotion Administration, Ministry of Health and Welfare, Taiwan. ^c^ Chronic disease included diabetes, heart disease, hypertension, stroke, arthritis, asthma, tuberculosis, peptic ulcer disease, chronic hepatitis and liver cirrhosis.

**Table 2 cancers-16-01217-t002:** Mean radiation dose (cGy) of VMAT and IMPT on the organs at risk in nasopharyngeal cancer patients.

Organs at Risk	VMAT	IMPT	*p*-Value
	*n* = 246	*n* = 41	
Brain stem	3068.5 ± 407.9	2164.7 ± 540.8	<0.001
Larynx	3673.7 ± 954.3	1403.5 ± 439.8	<0.001
Right cochlea	4156.8 ± 1040.0	3203.1 ± 823.4	<0.001
Left cochlea	4152.8 ± 899.5	3674.3 ± 1025.0	0.039
Right eye	860.2 ± 283.1	618.8 ± 245.6	<0.001
Left eye	940.2 ± 552.5	627.1 ± 270.0	0.002
Right lens	608.8 ± 113.7	294.4 ± 152.6	<0.001
Left lens	577.0 ± 91.6	285.1 ± 160.4	<0.001
Right optic nerve	3128.1 ± 816.0	3118.4 ± 918.6	0.961
Left optic nerve	3058.0 ± 810.2	3099.5 ± 1087.8	0.849
Right parotid	3003.2 ± 399.5	2969.2 ± 396.1	0.703
Left parotid	2992.5 ± 565.8	2933.3 ± 309.6	0.560
Oral cavity	3628.1 ± 364.8	936.0 ± 360.9	<0.001
Thyroid	4165.2 ± 675.7	3694.0 ± 275.0	<0.001
Cervical esophagus	3493.2 ± 334.8	2684.9 ± 552.8	<0.001
Mandible	3898.9 ± 720.8	1735.9 ± 507.9	<0.001

VMAT: volumetric modulated arc therapy; IMPT: intensity-modulated proton therapy.

**Table 3 cancers-16-01217-t003:** Adjusted quality of life mean scores of EORTC QLQ-C30 scales at different time points of treatment by VMAT and IMPT.

Quality of Life	Pre-RT			During RT			3 Months Post RT			12 Months Post RT		
	VMAT	IMPT	Diff. ^b^	VMAT	IMPT	Diff. ^b^	VMAT	IMPT	Diff. ^b^	VMAT	IMPT	Diff. ^b^
	aMean (SE) ^a^	aMean (SE) ^a^		aMean (SE) ^a^	aMean (SE) ^a^		aMean (SE) ^a^	aMean (SE) ^a^		aMean (SE) ^a^	aMean (SE) ^a^	
**Global QoL**	54.4 (1.4)	61.5 (3.5)	7.2	38.5 (1.3)	53.1 (3.2)	14.6 *	60.8 (1.4)	64.3 (3.3)	3.5	68.4 (1.3)	69.2 (3.0)	0.8
**Functional QoL**												
Physical	92.1 (0.8)	91.0 (2.1)	−1.1	82.3 (1.1)	90.3 (2.6)	8.0 *	88.0 (0.9)	91.7 (2.1)	3.7	91.3 (0.7)	96.6 (1.6)	5.3 *
Role	89.3 (1.3)	82.1 (3.4)	−7.2	73.9 (1.9)	82.5 (4.5)	8.6	87.9 (1.3)	85.0 (3.1)	−2.9	92.8 (1.1)	96.6 (2.5)	3.8
Emotional	76.1 (1.2)	81.2 (2.9)	5.1	75.3 (1.4)	79.5 (3.3)	4.2	84.9 (1.1)	89.2 (2.8)	4.3	85.1 (1.2)	91.8 (2.6)	6.7 *
Cognitive	86.0 (1.0)	92.0 (2.4)	6.0 *	80.3 (1.3)	88.1 (3.2)	7.8 *	84.7 (1.2)	88.5 (2.9)	3.8	83.7 (1.1)	94.0 (3.3)	10.3 *
Social	74.1 (1.5)	78.2 (3.7)	4.1	65.9 (1.7)	75.6 (4.0)	9.7 *	77.8 (1.5)	85.3 (3.6)	7.5	82.9 (1.3)	92.6 (2.9)	9.7 *
**C30 symptoms**												
Fatigue	22.4 (1.3)	24.5 (3.2)	2.1	43.5 (1.4)	40.8 (3.5)	−2.7	28.5 (1.3)	23.3 (3.1)	−5.2	23.5 (1.3)	13.3 (2.9)	−10.2 *
Nausea and vomiting	9.5 (1.2)	14.6 (3.0)	5.1	37.7 (1.8)	30.3 (4.4)	−7.4	9.5 (1.1)	6.3 (2.8)	−3.2	4.9 (0.7)	0.4 (1.6)	−4.5 *
Pain	15.5 (1.3)	12.3 (3.1)	−3.2	35.2 (1.7)	24.9 (4.0)	−10.3 *	14.0 (1.3)	9.7 (3.1)	−4.3	12.6 (1.2)	9.2 (2.8)	−3.4
Dyspnea	7.9 (1.0)	8.9 (2.4)	1.0	12.9 (1.3)	9.8 (3.1)	−3.1	8.4 (1.1)	5.7 (2.7)	−2.7	8.2 (1.1)	4.9 (2.5)	−3.3
Insomnia	25.7 (1.6)	20.9 (4.1)	−4.8	29.8 (1.8)	33.3 (4.3)	3.5	22.3 (1.6)	15.8 (3.9)	−6.5	22.9 (1.7)	20.8 (4.0)	−2.1
Appetite loss	17.0 (1.6)	15.9 (4.0)	−1.1	57.5 (1.9)	54.7 (4.6)	−2.8	27.1 (1.7)	18.2 (4.2)	−8.9	13.2 (1.5)	7.0 (3.3)	−6.2
Constipation	12.5 (1.3)	14.1 (3.3)	1.6	26.2 (1.7)	19.4 (4.1)	−6.8	16.3 (1.5)	11.9 (3.7)	−4.4	13.5 (1.4)	7.9 (3.2)	−5.6
Diarrhea	11.0 (1.0)	5.7 (2.6)	−5.3	13.1 (1.3)	9.8 (3.2)	−3.3	10.1 (1.2)	2.5 (2.9)	−7.6 *	7.4 (1.0)	2.8 (2.2)	−4.6
Financial difficulties	25.3 (1.6)	19.2 (4.1)	−6.1	23.7 (1.7)	7.9 (4.0)	−15.8 *	21.2 (1.6)	4.6 (3.9)	−16.6 *	17.9 (1.6)	3.0 (3.6)	−14.9 *

RT: radiation therapy; VMAT: volumetric modulated arc therapy; IMPT: intensity-modulated proton therapy; SE, standard error; *: *p* < 0.05. ^a^ Adjusted means (aMean) were obtained from a linear regression model adjusted for age, sex, ethnicity, education, body mass index, chronic diseases, AJCC stage, and chemotherapy. ^b^ Diff. represents the score difference between the IMPT group and the VMAT group.

**Table 4 cancers-16-01217-t004:** Adjusted quality of life mean scores of EORTC QLQ-HN35 scales at different time points of treatment by VMAT and IMPT.

Quality of Life	Pre-RT			During RT			3 Months Post RT			12 Months Post RT		
	VMAT	IMPT	Diff. ^b^	VMAT	IMPT	Diff. ^b^	VMAT	IMPT	Diff. ^b^	VMAT	IMPT	Diff. ^b^
	aMean (SE) ^a^	aMean (SE) ^a^		aMean (SE) ^a^	aMean (SE) ^a^		aMean (SE) ^a^	aMean (SE)^a^		aMean (SE) ^a^	aMean (SE) ^a^	
**HN35 symptoms**												
Pain	7.3 (0.7)	1.1 (1.9)	−6.2 *	37.1 (1.6)	16.8 (3.8)	−20.3 *	15.5 (1.2)	6.2 (2.9)	−9.3 *	9.3 (0.8)	2.0 (1.9)	−7.3 *
Swallowing	6.9 (0.8)	3.5 (2.1)	−3.4	37.6 (1.6)	20.3 (3.9)	−17.3 *	18.8 (1.2)	10.3 (3.0)	−8.5 *	14.1 (1.1)	5.2 (2.5)	−8.9 *
Senses problems	8.1 (1.0)	6.2 (2.5)	−1.9	49.2 (1.6)	44.9 (3.9)	−4.3	25.8 (1.5)	16.1 (3.7)	−9.7 *	18.6 (1.4)	5.2 (3.3)	−13.4 *
Speech problems	7.3 (0.8)	5.0 (2.0)	−2.3	22.8 (1.5)	13.2 (3.7)	−9.6 *	16.6 (1.2)	11.6 (3.0)	−5.0	11.8 (1.0)	6.0 (2.3)	−5.8 *
Trouble with social eating	6.9 (0.9)	7.8 (2.3)	0.9	44.9 (1.7)	34.7 (4.1)	−10.2 *	19.9 (1.3)	10.6 (3.3)	−9.3 *	11.3 (1.1)	4.0 (2.4)	−7.3 *
Trouble with social contact	5.2 (0.7)	6.3 (1.7)	1.1	19.1 (1.3)	12.8 (3.3)	−6.3	10.8 (1.0)	4.8 (2.6)	−6.0 *	6.2 (0.8)	3.9 (1.8)	−2.3
Less sexuality	15.0 (1.4)	19.5 (3.4)	4.5	36.7 (2.3)	31.0 (5.6)	−5.7	26.2 (1.8)	18.3 (4.4)	−7.9	23.1 (1.7)	7.8 (3.9)	−15.3 *
Teeth	22.5 (1.5)	7.7 (3.7)	−14.8 *	21.7 (1.5)	10.9 (3.6)	−10.8 *	25.0 (1.6)	7.2 (4.0)	−17.8 *	26.5 (1.7)	8.7 (3.9)	−17.8 *
Opening mouth	5.4 (0.9)	2.3 (2.2)	−3.1	20.8 (1.5)	5.0 (3.6)	−15.8 *	14.4 (1.4)	8.2 (3.5)	−6.2	13.5 (1.4)	8.0 (3.2)	−5.5
Dry mouth	22.5 (1.5)	8.4 (3.7)	−14.1 *	60.5 (1.7)	66.4 (4.1)	5.9	56.8 (1.7)	60.3 (4.1)	3.5	47.8 (1.9)	37.6 (4.3)	−10.2 *
Sticky saliva	16.4 (1.4)	4.6 (3.5)	−11.8 *	60.1 (2.0)	62.3 (4.8)	2.2	42.4 (1.9)	58.8 (4.6)	16.4 *	32.4 (1.9)	41.4 (4.3)	9.0
Coughing	19.8 (1.3)	10.3 (3.1)	−9.5 *	35.9 (1.9)	22.0 (4.5)	−13.9 *	21.2 (1.4)	11.2 (3.4)	−10.0 *	21.8 (1.6)	8.8 (3.7)	−13.0 *
Felt ill	22.4 (1.5)	20.3 (3.7)	−2.1	48.7 (1.9)	39.2 (4.6)	−9.5	27.2 (1.5)	15.5 (3.7)	−11.7 *	18.8 (1.5)	11.3 (3.4)	−7.5 *
Pain killers	38.6 (3.1)	20.8 (7.7)	−17.8 *	51.4 (3.4)	28.5 (8.2)	−22.9 *	16.0 (2.5)	11.4 (6.2)	−4.6	16.3 (2.7)	13.7 (6.2)	−2.6
Nutritional supplements	38.3 (3.1)	58.1 (7.9)	19.8 *	82.3 (2.8)	69.9 (6.7)	−12.4	46.3 (3.5)	40.3 (8.7)	−6.0	31.1 (3.4)	32.6 (7.7)	1.5
Feeding tube	1.4 (0.7)	0.4 (1.7)	−1.0	5.8 (1.5)	0.9 (3.7)	−4.9	4.2 (1.3)	1.3 (3.3)	−2.9	1.5 (0.9)	0.8 (1.9)	−0.7
Weight loss	33.6 (2.9)	12.2 (7.3)	−21.4 *	83.7 (2.6)	50.0 (6.4)	−33.7 *	49.7 (3.6)	42.9 (8.8)	−6.8	26.1 (3.2)	16.3 (7.2)	−9.8
Weight gain	17.6 (2.6)	41.3 (6.4)	23.7 *	10.8 (2.1)	12.2 (5.2)	1.4	31.7 (3.2)	15.0 (7.8)	−16.7	52.9 (3.6)	33.9 (8.3)	−19.0 *

RT: radiation therapy; VMAT: volumetric modulated arc therapy; IMPT: intensity-modulated proton therapy; SE, standard error; *: *p* < 0.05. ^a^ Adjusted means (aMean) were obtained from a linear regression model adjusted for age, sex, ethnicity, education, body mass index, chronic diseases, AJCC stage, and chemotherapy. ^b^ Diff. represents the score difference between the IMPT group and the VMAT group.

**Table 5 cancers-16-01217-t005:** Group and time effects of RT technique and time points on average scores for EORTC QLQ-C30 and QLQ-HN35 quality of life outcomes.

Factor	Global QoL			Functional QoL ^a^			C30 Symptoms ^a^			HN35 Symptoms ^a^		
	adj. β ^b^	(95% CI)	*p*-Value	adj. β ^b^	(95% CI)	*p*-Value	adj. β ^b^	(95% CI)	*p*-Value	adj. β ^b^	(95% CI)	*p*-Value
**Group effect**												
Group												
VMAT	Ref.			Ref.			Ref.			Ref.		
IMPT	7.2	(2.4, 11.9)	0.003	2.5	(−2.4, 7.4)	0.319	−4.7	(−7.8, −1.5)	0.003	−3.6	(−8.1, 0.9)	0.121
**Time effect**												
Time points												
Pre-RT	Ref.			Ref.			Ref.			Ref.		
During RT	−15.3	(−18.5, −12.2)	<0.001	−8.4	(−10.3, −6.6)	<0.001	14.5	(12.8, 16.3)	<0.001	24.6	(22.7, 26.4)	<0.001
3 months post RT	5.5	(1.8, 9.1)	0.003	1.2	(−1.1, 3.5)	0.307	0.6	(−1.5, 2.7)	0.582	9.7	(7.5, 12.0)	<0.001
1 year post RT	13.8	(9.9, 17.7)	<0.001	3.7	(1.0, 6.3)	0.006	−2.9	(−5.2, −0.6)	0.013	5.5	(3.0, 8.0)	<0.001
**Interaction effect** (Group × time points)												
VMAT × Pre-RT	N/A			Ref.			N/A			Ref.		
IMPT × During RT				5.0	(0.2, 9.8)	0.040				−7.1	(−12.0, −2.3)	0.004
IMPT × 3 months post RT				1.8	(−4.1, 7.6)	0.557				−1.4	(−7.1, 4.3)	0.626
IMPT × 12 months post RT				4.7	(−1.8, 11.1)	0.156				−3.0	(−9.2, 3.1)	0.326

RT, radiotherapy; VMAT: volumetric modulated arc therapy; IMPT: intensity-modulated proton therapy; Ref: reference group; N/A: not applicable since the interaction is not significant. ^a^ The scores were the quality of life average scores obtained from EORTC QLQ-C30 5 functional scales, 9 C30 symptoms, and 18 QLQ-HN35 symptoms or items. ^b^ Adjusted regression coefficients (adj. β) were obtained from generalized estimating equations adjusted for age, sex, ethnicity, education, body mass index, chronic diseases, AJCC stage, and chemotherapy.

## Data Availability

The data presented in this study are available from the corresponding author upon request. The data are not publicly available due to ethical restrictions.

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
