# Peer review of "Longitudinal Assessment of Quality of Life in Nasopharyngeal Cancer Patients Treated with Intensity-Modulated Proton Therapy and Volumetric Modulated Arc Therapy at Different Time Points"

_cancers, 2024, doi:10.3390/cancers16061217_

Round 1

Reviewer 1 Report

Comments and Suggestions for Authors

The investigated topic is intersting to radiooncologist dealing with ENT radiotherapy. Unfortunately, IMPT is not in wide use because of the metioned reason. Interestinglingy the authors showed, IMPT is not the source for solution of all problems / symptoms and impairmet of QoL NPC patients suffer from after radiochemotherapy. There is a clear statement given.

It seems there is a bias concerning the use of IMPT. Thus it is hardly to understand why the cognitive function of VMAT patients was lower before any treatment compared to IMPT patients. This should be a little bit more elucidated and explained.

I recommend to switch the first sentence of the "Results" in the abstract into another position. The main object of the investigation was the QoL and not the dose sparing opportunities of IMPT to the OARs, radiooncologists don´t expect anything else. 

Author Response

Reviewer 1

The investigated topic is interesting to radiooncologist dealing with ENT radiotherapy. Unfortunately, IMPT is not in wide use because of the mentioned reason. Interestingly the authors showed, IMPT is not the source for solution of all problems / symptoms and impairment of QoL NPC patients suffer from after radiochemotherapy. There is a clear statement given.

Response: Thank the reviewer for the comments.

It seems there is a bias concerning the use of IMPT. Thus it is hardly to understand why the cognitive function of VMAT patients was lower before any treatment compared to IMPT patients. This should be a little bit more elucidated and explained.

Response: Thank the reviewer for the comment. In Taiwan, IMPT treatment is not covered by insurance, so NPC patients who opt for IMPT therapy generally have higher socioeconomic status (SES) compared to those undergoing VMAT treatment. Previous studies have shown that higher SES is associated with better cognitive function [1-2]. This finding may help explain the results observed in our study. To address this potential confounding factor, we accounted for education level, which is a proxy for SES, in all our assessments. We have acknowledged this limitation in our manuscript on lines 403-408.

References

  1. Yang, L.; Martikainen, P.; Silventoinen, K.; Konttinen, H. Association of socioeconomic status and cognitive functioning change among elderly Chinese people. Age and Ageing 2016, 45, 674-680, doi:10.1093/ageing/afw107.
  2. Shi, L.; Tao, L.; Chen, N.; Liang, H. Relationship between socioeconomic status and cognitive ability among Chinese older adults: the moderating role of social support. Int J Equity Health 2023, 22, 70, doi:10.1186/s12939-023-01887-6.

I recommend to switch the first sentence of the "Results" in the abstract into another position. The main object of the investigation was the QoL and not the dose sparing opportunities of IMPT to the OARs, radiooncologists don´t expect anything else. 

Response: Thank you for the suggestion. In this manuscript, we aimed to showcase the potential relationship between dose reduction in specific organs at risk in IMPT treatments, and the subsequent improvement in QoL. To maintain the logical flow, we initially presented the dose data, followed by the findings regarding QoL. Therefore, we kindly request that the sequence of the findings displayed in the Abstract be preserved.

Reviewer 2 Report

Comments and Suggestions for Authors

This manuscript compares QoL in non-metastatic NPC treated with VMAT and IMPT (an important question) and reports a time-dependent improvement in functional QoL with the latter, attributed due to a significant dose reduction in OARs with protons relative to VMAT. On the whole, the manuscript is well written, with a good discussion. The authors recognise the limitation of a small cohort of IMPT patients, which makes making definitive conclusions challenging in this analysis.

There are few aspects, where the manuscript needs to be improved prior to publication

1. At different lines (36, 87, 163 for instance) - this work has been presented as a 'study'. It is unclear in the manuscript whether this was a prospective study or a retrospective institution-approved (more likely) analysis. This needs to be clarified, along with any ethics approval. Along the same lines, page 163 defines the pre-RT period as the time from when a patient agreed to participate in the study...... If this is a retrospective analysis, patients would not have been consented for this at the time - would be helpful to clarify please.

2. Study design : Was this analysis planned as a matched case-control study, or as a real-world data analysis? If former, how was the patient cohort size decided?

3. Patient cohort: line 91 - inclusion criteria included only those who completed the prescribed QoL questionnaires. Was that for each patient at all defined time-points - baseline, during RT, 3 and 12 months post treatment? Was there a 100 % return rate therefore for all 287 patients? There was no table stating compliance rates with QoL questionnaires. 

4. Co-variates: NPC was staged using AJCC 8th edition. It would be useful to clarify whether the authors restaged the NPC patients prior to  implementation of 8th edition in this patient cohort, as patients treated in 2011 would have been staged with older editions.

5. Table 1: Please add how many patients in each arm had induction chemotherapy (+ type) + CRT, versus CRT alone. Relevant given that diarrhoea has been marked as significant which could be a chemo tox (5-FU) rather than RT 

6. Baseline tumour characteristics : Was there any difference between the 2 groups in terms of GTV, CTV-H, and PTV volumes that could impact on dose differences and ultimately toxicity. For the IMPT group, were they actually planned to PTV or CTV with robust optimisation (and therefore smaller volume). Were there any difference over time in the institution with regards to prophylactic nodal irradiation when IMPT was adopted in 2019 (upper neck RT alone, omission of irradiation in node-neg neck for instance) that was not standard of care in patients treated with VMAT between 2011 and 2019. Also, was there any difference in feeding tube policy (reactive v prophylactic) that may have impacted on weight loss etc?

Author Response

Reviewer 2

This manuscript compares QoL in non-metastatic NPC treated with VMAT and IMPT (an important question) and reports a time-dependent improvement in functional QoL with the latter, attributed due to a significant dose reduction in OARs with protons relative to VMAT. On the whole, the manuscript is well written, with a good discussion. The authors recognise the limitation of a small cohort of IMPT patients, which makes making definitive conclusions challenging in this analysis.

Response: Thank the reviewer for the comments.

There are few aspects, where the manuscript needs to be improved prior to publication

  1. At different lines (36, 87, 163 for instance) - this work has been presented as a 'study'. It is unclear in the manuscript whether this was a prospective study or a retrospective institution-approved (more likely) analysis. This needs to be clarified, along with any ethics approval. Along the same lines, page 163 defines the pre-RT period as the time from when a patient agreed to participate in the study...... If this is a retrospective analysis, patients would not have been consented for this at the time - would be helpful to clarify please.

Response: Our study was a prospective cohort study conducted at the Department of Radiation Oncology in Kaohsiung Chang Gung Memorial Hospital. NPC patients were recruited from outpatient visits, where the treatment plan and radiotherapy schedule were determined. After obtaining informed consent from willing participants, the first QoL investigation was conducted prior to radiotherapy. On average, there was a waiting period of approximately 1 to 2 weeks between the confirmation of the treatment modality and the start of radiotherapy, referred to as the “Pre-RT period”. We appreciate the reviewer’s feedback and have included the relevant information and ethics approval statement in the revised manuscript. For more detailed information, please refer to the highlights provided in lines 86-94 and 167-169.

  1. Study design : Was this analysis planned as a matched case-control study, or as a real-world data analysis? If former, how was the patient cohort size decided?

Response: Our study was a real-world, longitudinal, concurrent prospective study. We appreciate the comments provided.

  1. Patient cohort: line 91 - inclusion criteria included only those who completed the prescribed QoL questionnaires. Was that for each patient at all defined time-points - baseline, during RT, 3 and 12 months post treatment? Was there a 100 % return rate therefore for all 287 patients? There was no table stating compliance rates with QoL questionnaires.

Response: Thank you for your comment. The return rates for the QoL investigation at the four time points were as follows: 100% (n = 287) for pre-RT, 86.8% (n = 249) during RT, 84.7% (n = 243) at 3 months post-RT, and 73.9% (n = 212) at 1 year post-RT. The relevant data has been included in the revised manuscript. Please refer to the lines 171-175 for more information.

  1. Co-variates: NPC was staged using AJCC 8th edition. It would be useful to clarify whether the authors restaged the NPC patients prior to implementation of 8th edition in this patient cohort, as patients treated in 2011 would have been staged with older editions.

Response: Thank you for your valuable comment. To ensure consistency, we restaged all 287 patients in this study using the criteria outlined in the AJCC 8th edition. We have revised the diagnosis statement in the revised manuscript. For more detailed information, please refer to lines 184-185.

  1. Table 1: Please add how many patients in each arm had induction chemotherapy (+ type) + CRT, versus CRT alone. Relevant given that diarrhoea has been marked as significant which could be a chemo tox (5-FU) rather than RT 

Response: Thank you for your valuable comment. We have revised the categorization of the variable "chemotherapy" into three distinct groups: "no" "concurrent," and "induction + concurrent." Upon analysis, no statistically significant difference was observed between the VMAT and IMPT groups for this variable, with a p-value of 0.300.

  1. Baseline tumour characteristics: Was there any difference between the 2 groups in terms of GTV, CTV-H, and PTV volumes that could impact on dose differences and ultimately toxicity. For the IMPT group, were they actually planned to PTV or CTV with robust optimisation (and therefore smaller volume). Were there any difference over time in the institution with regards to prophylactic nodal irradiation when IMPT was adopted in 2019 (upper neck RT alone, omission of irradiation in node-neg neck for instance) that was not standard of care in patients treated with VMAT between 2011 and 2019. Also, was there any difference in feeding tube policy (reactive v prophylactic) that may have impacted on weight loss etc?

Response: Thank you for your valuable comment. As indicated in the manuscript, target delineation, dose prescription, and fractionation protocols were consistent for both VMAT and IMPT patients (line 100-101). For the IMPT group, robust optimization techniques were employed to account for range uncertainties (plus 3.5%) and positional uncertainties (plus 3 mm). Please refer to the lines 117-119. Conversely, for the VMAT group, the planning target volume (PTV) was generated with additional margins of 3-5 mm around each clinical target volume (CTV). Please refer to the lines 124-125. Notably, prophylactic neck irradiation was not omitted for either group, even in cases with clinical stage T1N0. Furthermore, prophylactic tube feeding was not administered to patients in either group at our institute.

Reviewer 3 Report

Comments and Suggestions for Authors

The primary objective of this study was to assess the quality of life outcomes among nasopharyngeal cancer patients treated with either Intensity-Modulated Proton Therapy (IMPT) or Volumetric Modulated Arc Therapy (VMAT) at various stages of treatment. This research holds significant clinical relevance. The methodology employed was deemed appropriate. The study revealed that, in comparison to VMAT, IMPT led to a reduction in dosage which correlated with improved functional quality of life and reduced HN35 symptoms. However, this effect was found to be time-sensitive and was only evident during the radiation therapy phase. Furthermore, the manuscript demonstrates a high level of writing proficiency, with no apparent issues in English language usage. A minor concern arises in the "Methods" section, where it might be beneficial to provide a discussion on the methodology of generalized estimating equations. Since many readers may not be familiar with this analytical approach, elucidating its relationship to generalized linear models could enhance comprehension.

Author Response

Reviewer 3

The primary objective of this study was to assess the quality of life outcomes among nasopharyngeal cancer patients treated with either Intensity-Modulated Proton Therapy (IMPT) or Volumetric Modulated Arc Therapy (VMAT) at various stages of treatment. This research holds significant clinical relevance. The methodology employed was deemed appropriate. The study revealed that, in comparison to VMAT, IMPT led to a reduction in dosage which correlated with improved functional quality of life and reduced HN35 symptoms. However, this effect was found to be time-sensitive and was only evident during the radiation therapy phase. Furthermore, the manuscript demonstrates a high level of writing proficiency, with no apparent issues in English language usage. A minor concern arises in the "Methods" section, where it might be beneficial to provide a discussion on the methodology of generalized estimating equations. Since many readers may not be familiar with this analytical approach, elucidating its relationship to generalized linear models could enhance comprehension.

Response: Thank you for your valuable comment. We have included the relevant statement regarding GEE in the revised manuscript. For more detailed information, please refer to lines 200-205.

Round 2

Reviewer 2 Report

Comments and Suggestions for Authors

Many thanks for addressing comments. 

As this is reported as a prospective cohort study, rather than retrospective analysis, please clarify inclusion criteria lines 93-95. It mentions that only those patients who completed a course of VMAT/IMPT and those who completed the QoL questionnaires were included. If the patients were consented to this prospective cohort study prior to starting treatment, it would not be possible to establish whether they would proceed to complete the entire treatment. Is it the case that all NPC patients in the defined time period were consented for the cohort study, and of them 287 completed treatment and were included for analysis. If so, please provide the numbers who were consented for the cohort study. Please also expand on 'who completed the prescribed QoL questionnaires' - I think you meant baseline questionnaires as you do not have a 100 % return rates at all time points for all patients.

Could the authors also clarify the prospective cohort study aspect please? IMPT only started later (not at the same time as VMAT), therefore would not have been defined at the beginning of the study, or considered for comparison later. It is possible that the institution had an ethics-approved ?registration cohort study for including all NPC patients and this is an analysis from the same after IMPT started

Return rates 177-179 - please move that to results section

Author Response

As this is reported as a prospective cohort study, rather than retrospective analysis, please clarify inclusion criteria lines 93-95. It mentions that only those patients who completed a course of VMAT/IMPT and those who completed the QoL questionnaires were included. If the patients were consented to this prospective cohort study prior to starting treatment, it would not be possible to establish whether they would proceed to complete the entire treatment. Is it the case that all NPC patients in the defined time period were consented for the cohort study, and of them 287 completed treatment and were included for analysis. If so, please provide the numbers who were consented for the cohort study. Please also expand on 'who completed the prescribed QoL questionnaires' - I think you meant baseline questionnaires as you do not have a 100 % return rates at all time points for all patients.

Could the authors also clarify the prospective cohort study aspect please? IMPT only started later (not at the same time as VMAT), therefore would not have been defined at the beginning of the study, or considered for comparison later. It is possible that the institution had an ethics-approved registration cohort study for including all NPC patients and this is an analysis from the same after IMPT started.

Return rates 177-179 - please move that to results section

Response:

Thank you for your insightful feedback. We acknowledge our previous oversight in the understanding of our study design and have rectified it throughout the entire manuscript. Additionally, in accordance with your suggestion, we have relocated the sentences regarding return rates to the Results section.